# Elaboration of Thermally Performing Polyurethane Foams, Based on Biopolyols, with Thermal Insulating Applications

**DOI:** 10.3390/polym16020258

**Published:** 2024-01-16

**Authors:** Pedro Luis De Hoyos-Martinez, Sebastian Barriga Mendez, Eriz Corro Martinez, De-Yi Wang, Jalel Labidi

**Affiliations:** 1Chemical and Environmental Engineering Department, University of the Basque Country, Plaza Europa 1, 20018 Donostia-San Sebastián, Spain; sebastian.barriga@ehu.eus; 2Chemical and Environmental Engineering Department, University of the Basque Country, Otaola Etorbidea 29, 20600 Eibar, Spain; erizcm1990@gmail.com; 3IMDEA Materials Institute, C/Eric Kandel, 2, 28906 Getafe, Spain; deyi.wang@imdea.org; 4Escuela Politécnica Superior, Universidad Francisco de Vitoria, Ctra. Pozuelo-Majadahonda Km 1800, 28223 Pozuelo de Alarcón, Spain

**Keywords:** polyurethanes, biopolyol, fireproofing, inorganic fillers, thermal insulator

## Abstract

In this work, biobased rigid polyurethane foams (PUFs) were developed with the aim of achieving thermal and fireproofing properties that can compete with those of the commercially available products. First, the synthesis of a biopolyol from a wood residue by means of a scaled-up process with suitable yield and reaction conditions was carried out. This biopolyol was able to substitute completely the synthetic polyols that are typically employed within a polyurethane formulation. Different formulations were developed to assess the effect of two flame retardants, namely, polyhedral oligomeric silsesquioxane (POSS) and amino polyphosphate (APP), in terms of their thermal properties and degradation and their fireproofing mechanism. The structure and the thermal degradation of the different formulations was evaluated via Fourier Transformed Infrared Spectroscopy (FTIR) and thermogravimetric analysis (TGA). Likewise, the performance of the different PUF formulations was studied and compared to that of an industrial PUF. From these results, it can be highlighted that the addition of the flame retardants into the formulation showed an improvement in the results of the UL-94 vertical burning test and the LOI. Moreover, the fireproofing performance of the biobased formulations was comparable to that of the industrial one. In addition to that, it can be remarked that the biobased formulations displayed an excellent performance as thermal insulators (0.02371–0.02149 W·m^−1^·K^−1^), which was even slightly higher than that of the industrial one.

## 1. Introduction

Nowadays, polyurethanes (PUs) are classified as sixth among all the different polymers owing to their worldwide production and diverse number of applications [1]. According to a recent study, the global polyurethane market size was valued at USD 70.67 billion and nearly 24 million metric tons in 2020, and it is expected to grow at a compound annual growth rate (CAGR) of 3.8% in the next 7 years [2]. On the other hand, the polyurethane consumption was estimated to be about 6–8% of the total plastics during 2019, both in the United States of America and in Europe [3].

Considering their market applications, polyurethane foams (PUFs) represent the largest segment with a 65% share of the total market, followed by coatings (13%), elastomers (12%), adhesives (7%) and smart materials for the biomedical sector (3%) [4,5]. PUFs are mostly applied as insulators in the transportation and construction sectors. In this sense, they are the main thermal insulators that are employed in the market due to their low density and thermal conductivity compared to other materials such as mineral wool, polymers like polystyrene or lignocellulose products [6]. In general, polyurethane foams are classified into flexible and rigid foams [7]. The former cover half of the total worldwide production of PUFs, and they constitute the broadest part of the whole polyurethane family in terms of consumption [8].

Conventionally, the synthesis of polyurethanes is based on the reaction between polyols and diisocyanates to create urethane linkages. Consequently, their industrial synthesis is significantly dependent on petrochemicals. In recent decades, there has been an increasing awareness of the depletion of fossil resources, the environmental impact of petroleum-based products and the toxicity of raw materials [9]. In the field of polyurethanes, this has promoted the replacement of toxic and fossil-dependent components. In fact, by substituting these constituents with other ones of a more natural origin and renewable nature, it is possible to improve the environmental properties and to reduce the health-derived risks of the polyurethane foams [10]. With this aim, the main strategies implemented are the substitution of traditional isocyanates or the synthesis of new bioisocyanates and the replacement of synthetic polyols. The former methodology is based either on the search for alternatives to polyisocyanates, such as polyamines and polycyclic carbonates, which would be able to bring the same effect and properties to the foams [11] or on the synthesis of polyisocyanates from renewable raw materials, namely, sugars, oils or amino acids [12]. The latter approach involves the utilization of biobased polyols, such as those obtained from biomass, especially of the lignocellulosic type [13]. Considering the lower degree of development of the methods for producing isocyanates from agriculture and forestry industry wastes, research efforts have been mostly oriented toward the valorization of those raw materials towards the synthesis of biopolyols.

Nowadays, biomass and woody wastes represent a source of materials with a wide availability, low price and sustainable nature. Within the industry, though, these kinds of residues are generally disposed of or burned, with the corresponding environmental issues. Accordingly, current efforts are oriented towards a more efficient use of these feedstocks, which can have a positive impact on the environment [14]. This is of significant relevance in the case of lignocellulosic biomass derived from the forestry industry, whose biomass wastes are highly underutilized [15]. Moreover, this type of biomass represents a large source of natural hydroxyl groups, i.e., cellulose, hemicellulose and lignin, which are fundamental for the production of polyols [16]. At the present time, the conversion of this biomass into polyols by means of liquefaction is preferred, and therefore, it has been widely studied in recent years [17]. Liquefaction is a thermochemical process carried out with a solvent in acidic or alkaline media and with or without the presence of a catalyst at high temperatures, in which the components with a higher molecular weight of biomass are transformed into smaller fractions [18]. The solvents most commonly employed are polyhydric alcohols as polyethylene glycol (PEG), ethylene glycol (EG) and glycerol (G) [19]. Strong acids or bases are typically used as catalysts, especially sulfuric acid, which is able to reduce the temperature that is required for reaction [20]. Once the liquefaction is finished, the final product obtained is called biopolyol, and it is characterized by a high level of hydroxyl groups, derived from a mixture of carbohydrates, ethers, esters, glycols and acids [21].

Due to the large number of advantages of this methodology, a great number of works have appeared lately devoted to the production of biobased polyols via liquefaction for the synthesis of PUFs [22,23,24]. More specifically, several works can be found in the literature devoted to biopolyols produced by means of liquefaction from wood wastes, as this allows for their valorization. Nevertheless, in many of the works, there are still some inherent limitations to the process and to the later application. The scaling up of the liquefaction process represents one of the main current constraints. In this sense, several studies have been carried out on different wood wastes such *Eucalyptus globulus* or *Fagus sylvatica*, in which the amount of the solid that was fed into the reactor was lower than 10 g to obtain satisfactory results [19,25]. On the one hand, either high temperatures or considerably long times of reaction are generally employed to achieve significant yields. For instance, in some recent works using wood sawdust from *Alnus glutinosa*, it was presented that for achieving yields above 90%, temperatures of 170–180 °C and periods of even 6 h were required [15,26]. On the other hand, the ratios of liquid solvent to solid biomass are usually relatively high in order to result in a complete conversion of the wood waste. As an example, Olszeweski et al., 2023, displayed that it was possible to achieve the liquefaction of a cellulose sawdust waste with a yield of approximately 95%, but the ratio of solid to liquid had to be increased to 1:10 [27].

Considering the performance of the biopolyol-based PUFs, it is often below that of commercial PUFs, especially when there is a complete substitution of synthetic polyol by the biobased one. This can be observed especially regarding their thermal degradation and fireproofing properties. In fact, polyurethane foams are known as being highly flammable, which can lead to problems related to their toxicity and applications [28]. Moreover, flammability is reported to be an essential aspect of rigid polyurethane foams when used as thermal insulators [29]. For this reason, the employment of flame retardants and the study of their performance is often needed. In this respect, there has been a tendency towards the use of this type of additives with a greener nature. Accordingly, non-halogen compounds are selected nowadays, with special attention to phosphorous, nitrogenous and silicon compounds.

In this work, it was intended the valorization of wood sawdust, through the synthesis of biopolyols via liquefaction for the elaboration of PUFs. On the one hand, the efforts were oriented towards the scaling up of the liquefaction process. Consequently, the amount of biomass that was fed into the reactor was maximized, and the amount of solvent used was kept at a medium-low ratio. Moreover, the reaction conditions (temperature, time and amount of catalyst) were selected while aiming for a reduction in the inherent cost of the process. On the other hand, the elaboration of the PUFs was designed while focusing mainly on the thermal properties of the foams and the study of their fireproofing properties. Consequently, a sufficient degree of thermal resistance was aimed for. Thereby, two different halogen-free flame retardants were employed in the foam formulations to improve their fireproofing features. Lastly, all these properties of the biobased PUFs were compared to those of a commercial PUF to assess their performance.

## 2. Materials and Methods

### 2.1. Raw Materials and Reagents

The raw material used for the synthesis of the polyols was a residue from the wood industry, namely, sawdust from *Pinus radiate*, which was kindly provided by the company Ebaki XXI (Muxika, Spain). This raw material was characterized in terms of its chemical composition by using the corresponding TAPPI standard for the determination of extractives (T204-cm-97), ashes (T211 om-02) and lignin (T222-om-98). The contents of cellulose and hollocellulose were determined using the procedures described by Rowell 1984 [30] and Wise et al., 1946 [31], respectively.

During the process of liquefaction, the reagents employed were polyethylene glycol (PEG400) and sulfuric acid (96%), purchased from PanReac/AppliChem (Castellar del Vallès, Spain/Darmstadt, Germany), and glycerol from Sigma Aldrich (Darmstadt, Germany). Concerning the additives used for the preparation of the polyurethane foams, TEGOSTAB 84711 (EVONIK) was chosen as surfactant, dibutyltin dilaurate (DBTDL), delivered by Sigma Aldrich, and POLYCAT 5 by EVONIK (Polyurethane Foam Amine Catalyst Pentamethyldiethylenetriamine) were selected as catalysts, and water was utilized as foaming agent. Two flame retardants were employed, namely, SO1458-Trisilanol Phenyl POSS and EXOLIT AP422-ammonium polyphosphate, which were provided by Hybrid Plastics Inc. (Hattiesburg, MS, USA) and Clariant (Sulzbach am Taunus, Germany), respectively. As isocyanate Diphenyl, methane diisocyanate IsoPMDI 92149 was used, supplied by BASF Española S.L (Barcelona, Spain). An industrial polyol (IP), labelled as Poliuretan^®^Spray S-303HF from Synhtesia International S.L.U (Barcelona, Spain) and commercially available, was selected with comparison purposes to assess the properties of the biopolyol. This component was based on a mixture of polyols containing catalysts, flame-retardants and foaming agents (containing HFO).

Other reagents employed in the characterization analyses were NaOH and KOH (85%) in powder (PanReac/AppliChem), ethanol (Sharlau, Barcelona, Spain) and hydrochloric acid (37%) from Sigma Aldrich.

### 2.2. Synthesis of the Biopolyols: Liquefaction Process, Purification and Neutralization

The process for the synthesis of the biopolyols used a previous work that was carried out within our research group by Da Silva et al., 2019 [32], as a starting point. Nevertheless, in the current work, the aim was to scale up the process, and therefore, higher loadings of both raw material and reagents were employed.

Prior to the reaction, the sawdust was left under ambient conditions to release the excess of moisture. For the liquefaction, a glass triple neck round-bottom flask with a volume of 6 L was used as reactor, provided with a mechanical paddle blade stirrer, a thermocouple and a reflux condenser. Concerning the raw materials and reagents, 600 g of sawdust was fed into the reactor with 3 L mixture of PEG/G (60/40% *w*/*w*) as solvent (optimal ratio) and a 4.5% *w*/*w* (with respect to the solvent) of sulfuric acid as catalyst. The 600 g of sawdust was fed into a 6 L round-bottom glass reactor with 3 L of the solvent mixture PEG:G (ratio solid to liquid 1:5 *w*/*v*). The glass reactor was equipped with 4 openings for mechanical agitation, temperature probe, feed of the raw material and a condenser. First, the mixture of solvents and the catalyst were introduced into the reactor and heated until 70–80 °C. At this point, the sawdust was fed into the reactor little by little to avoid agglomeration. As the quantity of the raw material that was fed was increased, so was the temperature to promote good interaction between the solvents and catalyst and the sawdust. The reaction was run at 135 °C and 1 atm for 90 min. Once the liquefaction was finished, the obtained raw biopolyols had acetone added to them and were filtered with cellulose filters (12 µm) under vacuum to remove the insoluble solid residue. Finally, the remaining biopolyols were purified by means of rotatory evaporator to recover the acetone added previously, and they were neutralized to pH 6–7. The diagram of the process previously described can be found in Appendix A.

#### Optimization of the Liquefaction Process—Ratio of Solvents

The current work was based on a previous one in which Kraft lignin was used as raw material. However, here, sawdust coming from *Pinus radiate* was employed instead. Therefore, it was necessary to test different mixtures of solvents for liquefaction with the aim of obtaining a high conversion of the raw material and biopolyols with suitable properties. The solvent ratios assessed were 75:25, 70:30 and 60:40 PEG:G. The biopolyols obtained from these different mixtures of solvents, were analyzed for some physical chemical parameters, i.e., density, viscosity, hydroxyl index and for their structure and functionalities (size exclusion chromatography and Fourier Transformed Infrared Spectroscopy analysis). Finally, and after evaluation of the results, the optimal ratio was selected, based on the highest yield of liquefaction and the most convenient features of the polyol for the production of polyurethane foams. This optimal ratio was used at a larger scale, as presented previously.

### 2.3. Characterization of the Biopolyols

The biopolyols obtained from the process of liquefaction were analyzed based on different physical–chemical properties. Each of the analyses performed was carried out in triplicate.

The density of the biopolyols was measured at ambient temperature (20 °C), and it was determined gravimetrically by weighting a volumetric flask with a known volume (5 mL) filled with the sample.

Concerning the measurement of the apparent viscosity, a viscometer (Fungilab) equipped with an adapter (LCP) APM/B was employed. For the determination of this parameter, a volume of 6 mL of biopolyol, a rotational speed of 5 rpm (samples at ambient temperature) and of 20 rpm (samples at operation temperature), and a cylindrical spindle TL5 were selected. The viscometer was run for 1 h, and once the registered values were stabilized, the final apparent viscosity was obtained.

The determination of the hydroxyl value of the biopolyols was carried out in accordance with ASTM D4274-99: Standard Test Method for Testing Polyurethane Raw Materials: Determination of Hydroxyl Numbers of Polyols [33]. For the calculation of the mentioned parameter, Equation (1) was used:(1)$$I_{OH}=\frac{\left[\left(B-A\right)\cdot N\cdot 56.1\right]}{m_{polyol}}\cdot 100$$
where A represents the volume of solution required for titration of the sample (mL), B is the volume of solution required for titration of the blank (mL), N is the the normality of the solution used for the titration and $m_{polyol}$ represents the amount of sample used (g).

Taking into account the previous calculation, equivalent weight ($E_{polyol}$) was also determined by means of Equation (2), which takes into consideration the hydroxyl index that was previously obtained.
(2)$$E_{polyol}=\frac{1000\times 56.1}{I_{OH}}$$

The biopolyols were also analyzed for their molecular weight by means of high-performance size exclusion chromatography (HPSEC). A Jasco instrument was used, set up with a LC Net II/ADC interface, a reflex index detector RI-2031Plus and two PolarGel-M (300 mm × 7.5 mm) columns displayed in series. The mobile phase employed for the analyses was dimetylformamide with 0.1% lithium bromide, which was set at 40 °C with a flow of 0.7 mL∙min^−1^. The calibration of the apparatus was carried out previously with polystyrene standards ranging between 70,000 and 266 g mol^−1^.

The functionality of the biopolyols was calculated as well, based on the relationship between their molecular (M_n_) and equivalent weight ($E_{polyol}$) as shown in Equation (3).
(3)$$F=\frac{M_{n}}{E_{polyol}}$$

### 2.4. Preparation of the Polyurethane Foams

The elaboration of the polyurethane foams was carried out by means of a one-step two-phase method. For the formulations based on the biopolyols that were synthesized, the two phases employed were phase A, which consisted of the biopolyol, surfactant, catalyst, blowing agent and flame retardant and phase B, which was formed by diphenyl methane diisocyanate (IsoPMDI 92140). The subsequent process to produce the PUFs is described next. Briefly, the different additives were added to the biopolyol according to their corresponding ratios in a 500 mL plastic beaker. In the formulations with a flame retardant, this was added to the polyol prior to the other additives. Thus, it could be dispersed homogeneously into the biopolyol. However, in the case of POSS, for the proper incorporation into the polyol, a pretreatment was needed. First, the flame retardant was dissolved in ethanol (50% *w*/*w*), and then it was added to the polyol and mixed under strong mechanical agitation in an ultrasound bath.

When the polyol and the rest of the additives were incorporated, the mixture (phase A) was heated in a water bath at 45 °C. Simultaneously, the corresponding amount of IsoPMDI 92140 (phase B) was poured into 250 mL glass beaker and also heated up to 45 °C in the same water bath. When the desired temperature was reached, the mixture of biopolyol and additives (phase A) was subjected to mechanical agitation for 60 s to ensure the suitable dispersion of all components. Right after phase A was stirred for 60 s, phase B was added to the plastic beaker and both phases were agitated for 3–4 s (mixing time). After this short period, the mixture was poured into a rectangular silicon mold (170 × 90 × 64 mm) where the mixture was allowed to rise under ambient conditions. After 5 min, once the mold was cooled down, the foam was taken out, and it was left at room temperature (20–25 °C) for 36 h (curing time).

For the industrial formulation (F_IND), the polyurethane foams were produced from the mixture of two components, namely, Poliuretan^®^Spray S-303HFO (Synhtesia International S.L.U, Barcelona, Spain) and diphenyl methane disocyanate (IsoPMDI 92140, BASF Española S.L, Barcelona, Spain). The process of preparation of the PUFs was analogous to the one previously described. The only difference was that in this case, the former component (phase A) included the polyol and the different additives.

The diagram of the previously presented processes for the elaboration of biopolyol-based polyurethanes and those based on industrial polyols can be seen in Appendix A.

#### Development of the PUF Formulations: Optimization Process

In the selection of the different additives and the setting of their suitable ratios, a previous optimization was carried out. Thereby, for the development of PUF formulations, three stages were followed. First, the surfactant, catalyst and blowing agent had to be selected, and their optimal ratio in the foam formulations had to be set. Concerning the type of these additives, different types were considered at the beginning. As surfactants, TEGOSTAB B 84711, TEGOSTAB B 8871 and Silicone oil were tested, as catalysts, POLYCAT 5, POLYCAT 10 and Dibutyltin dilaurate (DBTDL) were assessed, and as blowing agents, water and pentane were studied. In respect to the surfactant and catalyst, TEGOSTAB B 84711, DBTDL and POLYCAT 5 were selected, as they were the only mixture that did not show problems of phase separation of the mixture after 72 h. Concerning the blowing agent, water was selected on two bases: On the one hand, it is a more environmentally friendly product without problems of volatility, unlike pentane. On the other hand, water allowed for a more controlled and adequate foaming reaction compared to pentane. Regarding the optimal ratio of these additives, since a great number of trials were conducted, it was not possible to measure all the properties for all the PUFs prepared. Consequently, the appearance in terms of rigidity, homogeneity, fragility or brittleness and the density were the points on which the focus was placed. In the end, the formulation in which TEGOSTAB B 84711, DBTDL, POLYCAT 5 and water were incorporated in 20%, 16%, 2% and 1% (*w*/*w*) with respect to the biopolyol was selected, owing to the strong rigidity, homogeneity and adequate dispersibility of the additives and suitable density. Then, the optimization was centered on the introduction of the flame retardants and the maximum loading possible. For this purpose, in the beginning, three different options were considered, i.e., ammonium polyphosphate (APP), polyhedral oligomeric silsesquioxane (POSS) and organically modified montmorillonite (OMMT). The latter one was dismissed due to solubility and dispersibility problems with the biopolyols, which resulted in PUFs with poor homogeneity. Different loadings were tested, and those which allowed for an adequate foaming process were tested regarding the standard for the analysis of the fire reaction for construction materials. Thereby, a 12.5% loading for APP and POSS was finally selected.

It must be highlighted that although in the previous optimizations, the NCO:OH index was fixed at 100 (which was the value used by our industrial partner), a third optimization was performed after the optimal value of all additives was found. This optimization was related to the NCO:OH index, and three ratios were tested, namely, 80, 120 and 160. Again after the evaluation of dispersion, homogeneity, rigidity, density and also the control of the foaming reaction, it was observed that an NCO:OH index of 120 resulted in the most favorable PUF formulations.

Consequently, all the polyurethane foam formulations (both those based on the biopolyol and that based on the industrial polyol) were prepared by using a ratio NCO:OH of 120. Thereby, it was assured that the same amount of phase A and B was employed in all the formulations. Concerning the biobased formulation of the polyurethane formulations, the ratio of the different components is shown in Table 1. With respect to the industrial formulation, the ratios and specific components could not be displayed in the above-mentioned table, owing to confidentiality requirements. In this case, as mentioned before, the formulations were mixed by using a ratio NCO:OH of 120 *w*/*w* (isocyanate: industrial polyol). For the calculation of the required amount of diiscyanate (IsoPMDI 92140), Equation (4) was employed:(4)$${NCO}_{index}=\frac{n_{iso}}{\sum \left(n_{polyol}+n_{H_{2}O}\right)}\cdot 100=\frac{\frac{m_{iso}}{E_{iso}}}{\frac{m_{polyol}}{E_{polyol}}+\frac{m_{H_{2}O}}{E_{H_{2}O}}}\cdot 100=\frac{m_{iso}\cdot \frac{NCO}{4202}}{m_{polyol}\cdot \frac{OH}{56100}+2\cdot \frac{m_{H_{2}O}}{18}}\cdot 100$$
where n_iso_ is the number of equivalents of isocyanate, n_polyol_ represents the number of equivalents of polyol, $n_{H_{2}O}$ represents the number of equivalents of water, m_iso_ is the weight of isocyanate, m_polyol_ is the weight of polyol, $m_{H_{2}O}$ is the weight of water, E_iso_ is the equivalent of isocyanate, E_polyol_ is the equivalent of polyol and $E_{H_{2}O}$ is the equivalent of water.

Here, the NCO index was already set (120); the mass of the polyol (defined as mass of industrial polyol or mass of mixture of biopolyol additives) was set; OH was the hydroxyl index of the mixture of biopolyol additives or that of the industrial polyol, and NCO was the content of isocyanate groups, which was a known parameter of IsoPMDI 92140 supplied by the industrial partner.

The elaboration of the polyurethane foams was carried out by means of a one-step two-phase method, with a ratio of NCO/OH of 120.

### 2.5. Characterization of the Polyurethane Foams

#### 2.5.1. Structural Analysis by Fourier Transformed Infrared Spectroscopy (FT-IR)

This test was carried out by using a Spectrum Two FTIR Spectrometer from Perkin Elmer (Shelton, CT, USA), which counted with an accessory L1050231 Universal for Attenuated Total Reflectance (ATR). The analysis of the samples was performed setting a number of 64 scans accumulated in transmission mode and a resolution of 4 cm^−1^. The spectrum range was fixed between 4000 and 400 cm^−1^.

#### 2.5.2. Thermogravimetric Analysis (TGA)

The thermal properties of the polyurethane foams prepared were analyzed via their corresponding thermogram (TG) and first derivative (DTC). The equipment employed was an RSI analyzer 851 from Mettler Toledo (Columbus, OH, USA). Samples in the range of 5–10 mg were analyzed between 25 and 800 °C with a heating rate of 10 °C∙min^−1^. An inert atmosphere was selected for the tests, with a flow of 50 mL∙min^−1^.

#### 2.5.3. Study of Polyurethane Foams Performance

In the assessment of the performance of the foams, different properties were analyzed. These are the properties that are typically examined in commercial polyurethane foams, which are produced industrially. For each of the analyses, the foam samples were previously cut to the appropriate dimensions according to the standard followed.

For the determination of the apparent density, the tests were carried out in accordance with UNE-EN 1602:2013 Standard: Thermal insulating products for building applications—Determination of the apparent density [34].

The measurement of the mechanical properties of the foams was based on UNE-EN 826:2013 Standard: Thermal insulating products for building applications—Determination of compression behavior [35].

For the determination of the surface morphology of the foams, scanning electron microscopy W filament (SEM) from JEOL JSM-6400 (Tokyo, Japan) and a resolution of 3.5 nm (in secondary electron mode and 30 kV) was employed. Prior to the analysis, the surface of the samples was coated by means of a vacuum sputter from EMITECH K550X (Paris, France) (0.1 mbar, 25 mA and 3 min). Then, the gold-coated samples were analyzed under 10 kV acceleration voltage, and images were taken with 30 and 60 times magnification. The average pore size of the foams was determined from the diameter estimate of 100 cells from each image, which was analyzed with a processing image software.

For the assessment of the fireproofing properties of these materials, three different methods were employed. First, the polyurethanes were examined based on UNE-EN ISO 11925-2:2021 Standard: Reaction to fire tests—Ignitability of products subjected to direct impingement of flame—Part 2: Single-flame source test. Secondly, [36], UL-94 vertical burning tests was performed in accordance with the ASTM D3801-10 Standard Test Method for Measuring the Comparative Burning Characteristics of Solid Plastics in a Vertical Position. In addition to those [37], limited oxygen index (LOI) test was carried out in compliance with UNE-EN ISO 4589-2:2017 Standard Determination of Fire behavior by means of limit oxygen index—Part 2: Tests at ambient temperature [38].

The efficiency achieved by the polyurethane foams as insulators was analyzed with regard to the UNE-EN 12667:2002 Standard: Thermal performance of building materials and products. Determination of thermal resistance was achieved by means of guarded hot plate and heat flow meter methods. Products were of high and medium thermal resistance [39]. This test was performed with a conductimeter instrument from Hot Disk TPS-1500 (Göteborg, Sweden) and a probe Kapton ref 5456 with a 3.189 mm radius.

## 3. Results and Discussion

### 3.1. Characterization of Raw Material and Liquefaction Process

The raw material that was employed for the liquefaction process, namely, pine wood sawdust, was analyzed in terms of its chemical composition (Table 2). The yield of the process and residue remaining was calculated as well.

As seen from the previous table, the raw material employed in the liquefaction process mostly consisted of cellulose and lignin, which was convenient for the latter synthesis of the biopolyol, since they are polyhidroxylated natural polymers. The content of both components was similar to those obtained in other analyses of *Pinus radiate* that were found in the literature [40,41]. In addition, the number of inorganic compounds that were determined presented a minor value. The percentage of extractives that were obtained was a bit higher than expected compared to other works [42,43]. This fact could be related to the mixture of solvents used in the extraction of these components. Moreover, it can also be associated with the fact that the tree species of origin was more mature, and therefore, it had a higher amount of extractives available [44].

Concerning the yield of liquefaction, a high degree of conversion of the raw material to the polyol was achieved, 89.15 ± 3.68. This achieved yield from the liquefaction process was at a similar level to or even slightly higher than those found in the literature [15,19,45]. Nevertheless, in this case, the variables of temperature and time were set at reduced values (135 °C, 90 min) compared to other works [46,47]. Moreover, in this work, the amount of solvent used was kept at a low ratio (1:5) compared to the works from the literature [29,48]. It should also be remarked that the liquefaction reaction was carried out successfully with an amount of solid fed into the reactor of 600 g sawdust. This was significantly higher than the amounts typically employed in the liquefaction of biomass, as reported by several authors [49,50,51]. In fact, it has been shown that when the amount of solid added to a reaction was increased, the yields of conversion were typically compromised [52]. Therefore, this point has usually remained a constraint on the industrial production of biopolyols from biomass. Nonetheless, in our case, only an amount of 65.09 ± 5.88 g of insoluble solid from the raw material remained after the liquefaction. Moreover, the final volume of polyol obtained reached 3.2 ± 0.28 L. Taking these points into consideration, it can be seen that a potential approximation to the scaling up of biopolyol production was accomplished.

### 3.2. Chemical and Structural Characterization of the Polyol

In this section, first, the results of the different polyols that were obtained in the optimization of the liquefaction process are shown. Then, the main properties of the selected formulation of the biopolyol that was synthesized from the liquefaction process are compared to those of the industrial polyol provided by the industrial partner.

#### 3.2.1. Structure and Performance of the Polyol during the Optimization of the Liquefaction Process

Here, the biopolyols that were obtained from the liquefaction process by using different ratios of the solvents (PEG:G) are shown. First, in Table 3, the main physical and chemical parameters are presented.

As seen from these results, in essential aspects, such as the hydroxyl index of the polyol, the functionality and the yield of conversion of the liquefaction, the polyols synthesized with a PEG:G ratio of 60:40 displayed the best results.

Concerning more structural parameters, such as the molecular weights shown in Table 4 and Figure 1, a comparison can be observed between the different biopolyols produced and the industrial polyol as well.

Here, it can be observed that as the amount of glycerol was becoming higher, the molecular weights were reduced, as was expected (a lower molecular weight of glycerol compared to polyethylene glycol). The use of polyols with a lower molecular weight is preferred to achieve better control of the chemistry and after-reaction with the isocyanate, which can promote a better foaming process.

The different biopolyols that were obtained were also assessed and compared between each other and with the industrial polyol regarding their functional groups by means of FTIR analysis. The spectra are presented in Figure 2.

From the spectra, it can be seen that there were not significant differences regarding the main functional groups of the biopolyols. A broad signal at 3400 cm^−1^ associated with the total hydroxyl groups was detected in all biopolyol formulations. Then, between 2800 and 300 cm^−1^, a sharper peak was seen related to the C-H of aliphatic methyl and methylene groups [15]. At 1700 cm^−1^, a small signal was observed, typically from the C=O of carbonyl groups. Another significant band was present in the range of 1200–1000 cm^−1^, which was associated with the C-O of eter and aliphatic primary and secondary alcohols [53]. Here, a noticeable difference was found between BP 60:40 and the other biopolyols. In the case of the former biopolyol, two peaks were detected at 1100 cm^−1^ and 1030 cm^−1^ (the latter was related to the deformation of primary alcohol [54], whereas in the other two, only one peak could be seen at 1100 cm^−1^. Thereby, the band of BP-60:40 was more similar to that of the industrial polyol than to the other two. In the final part of the spectra, small signals related to aromatic C=C and C-H linkages that are typical of a biomass were detected in all the biopolyols [55].

Considering all the previous information presented, it was found that the biopolyol formulation displaying the more convenient properties was the one obtained from the liquefaction process with a mixture of PEG:G (60:40). Consequently, this was the one selected and employed in the rest of the subsequent experiments.

#### 3.2.2. Assessment of the Optimized Biopolyol with the Industrial Polyol

Next, in Table 5, the main physical–chemical and structural parameters of the optimized biopolyol were analyzed and compared to those of the industrial polyol.

The biopolyol (BP) displayed slightly higher values of density and viscosity compared to those of the industrial polyol (IP). These values could be related to its biobased origin. Furthermore, the fact that after synthesis, the polyol was neutralized with NaOH in powder state might have also influenced this parameter. In any case, these values were lower than those presented by other authors [50,56]. This represented an advantage in the later production of polyurethanes in terms of processability.

The hydroxyl group index and functionality are prominent parameters for the polyols, with a direct influence on the polyurethane production. In the present work, both displayed higher values compared to the industrial polyol and well above 300 mg KOH∙g^−1^. Thus, the BP provided a major amount of OH that was available for the reaction with the isocyanate afterwards, which is of great convenience for the production of rigid PU foams [57]. Moreover, it would allow for a faster crosslinking for the production of the polyurethanes [58].

Concerning the structure of the polyols, the BP showed a higher molecular weight and size distribution. These characteristics were expected, and they were associated with the fact that BP was synthesized from a biomass by means of a liquefaction process. On the one hand, during the mentioned process, the BPs are synthesized via degradation of the large number of macromolecules of the lignocellulosic raw material [59]. Therefore, it is difficult to achieve molecular weights below 1000 Da, especially if a relatively high temperature or a high content of the acid catalyst are not employed. In addition, autocondensation reactions could occur, increasing the molecular weight values again [60]. On the other hand, in the production of synthetic polyols, the chemistry of the reaction is more easily controlled, since monomers and smaller molecules are used. For this reason, the size of the molecules that are synthesized is more uniform in the case of IPs compared to BPs with regard to the polydispersity index.

### 3.3. Characterization of the Foams

The different formulations of the synthesized PU foams were assessed regarding their structure by means of Fourier Transformed Infrared Spectra Analysis (FTIR) and concerning their thermal properties via thermogravimetric analysis (TGA).

It is shown in Figure 3a that the different formulations of PUFs that are synthesized from a biopolyol presented analogous signals to the industrial formulation. First, a peak from the stretching of N-H groups was detected at 3300 cm^−1^ [61]. This signal was within the band that typically covers the region of the O-H from hydroxyl groups. The band width was wider in the formulations that were synthesized from the biobased polyol. This effect was attributed to a higher content of hydroxyl groups within the BP compared to the IP. Following this peak, a band that can be attributed to the C-H bond of methyl and methylene groups was observed between 2900 and 2800 cm^−1^. Next, a sharp signal was seen at 1700 cm^−1^ related to carbonyl groups of the urethane linkages [22]. Another prominent band was obtained around 1500 cm^−1^, associated with the bending of N-H bonding [62]. Within this section of the spectra, other peaks of medium intensity were found at 1595 cm^−1^, 1414 cm^−1^ and 1215 cm^−1^, corresponding to the stretching of phenyl groups, isocyanurate rings and C-N stretching from urethane bonds [48,63,64]. At the end of the spectra, three peaks of moderate intensity were observed at 819 cm^−1^, 767 cm^−1^ and 515 cm^−1^, corresponding to the out-of-plan deformation of C-H linkages [65]. In addition to the previous signals, a significant band was observed at 1070–1000 cm^−1^. This is generally attributed to the C-O bonds of ether groups [66]. Nevertheless, considerable differences were observed between F_IND and the rest of the formulations. Those PUFs formulations that were derived from the BP displayed a sharper and larger band within this range. On the one hand, in F_REF, this was due to the presence of C-O from aliphatic primary and secondary alcohols, which increase the intensity of the signal. On the other hand, in the F_POSS and F_APP formulations, the additional presence of Si-O-Si and P-O-P signals also contributed to a larger band [67,68]. In the fingerprint part of the spectra (Figure 3b), low-intensity signals were also detected for those formulations. In the F_POSS formulation, they were associated with Si-C and Si-O at 695 cm^−1^ and 507 cm^−1^ [54], and in the F-APP formulation, a small peak was found at 884 cm^−1^, related to P-O-C [69].

Based on these prior observations, the typical reaction between the polyols and the isocyanate to yield urethane linkages and the presence of Si and P bonding derived from the use of POSS and APP, respectively, were confirmed in these formulations.

In Figure 4, the curves derived from the thermogravimetric analysis of the different PUF formulations are presented. Generally, polyurethanes display a thermal degradation that is constituted of various partial decomposition reactions [46]. In this sense, in Figure 4b, three major stages of degradation can be observed. First, between 180 and 265 °C, a medium-small step (5–10% weight) was noticed, which was related to the dissociation of urethane and urea bonds (rigid segments) [70]. Then, in the range of 275–360 °C, a more intense singnal (15–20% weight) was seen in all the formulations, for the decomposition of the polymeric chanis from the polyols. In the case of the biobased formulations, the peak can be more specifically related to the cellulose and lignin chais derived from the lignocellulosic biomass [71,72]. The final stage of degradation was detected around 380–465 °C, accompanied by another sharp and intense peak. This one could also be associated with the degradation of more condensed soft polyols segments and that of molecules originated from the combination of the polyol and the fireproofing fillers [73]. In general, all the formulations fit within these ranges. Nevertheless, it was seen that the F_REF showed the lowest temperatures of decomposition for each stage compared to the rest. This may be due to the absence of any flame retardant or fireproofing agent within the formulation. On the contrary, the highest temperatures of degradation of each stage were found for F_IND and F_APP, followed by F_POSS. This increase of the temperature was related to the presence of the different fireproofing additives.

From Figure 4a, different parameters were determined to assess further the thermal performance of the formulations, and these are gathered in Table 6. The onset temperature of degradation (T_5%_) did not show large differences between the formulations, except for the case of F_APP, for which the temperature was slightly higher and within the typical range for APP [74]. In fact, this formulation was the one that displayed the best thermal parameters. There, T_50%_ and the temperature of maximum degradation (T_max_) presented the highest values. Likewise, the percentage of residue remaining at the end was higher compared to the other formulations, whose values were within the same range. The reason for that could be the formation of a protective char layer during the degradation of APP [75]. The T_50%_ values were similar for F_IND and F_POSS, indicating that the addition of this filler was able to improve the thermal stability of the original biosourced PUF to the level of the industrial counterpart. The F_POSS formulation presented a higher temperature of maximum degradation as well compared to the industrial one. The value obtained rounded 400 °C, owing to the presence of POSS [76]. This confirmed the good dispersion and introduction of the filler into the PU system. Considering these results, it was corroborated that the introduction of the fireproofing additives into the PU formulations enhanced their resistance against thermal degradation.

### 3.4. Physical, Mechanical and Morphological Properties of the Foams

The different formulations of PUF elaborated were firstly analyzed for their density, compressive strength and morphology.

The apparent density of the PUF formulations was studied, as shown in Figure 5. In this figure, the effect of the different additives introduced into the formulations can be observed, especially compared to the industrial formulation. First, it was seen that the F_REF formulation displayed a slightly higher value of density compared to F_IND. This was directly related to the use of a polyol derived from the liquefaction of lignocellulosic biomass, which presented a higher hydroxyl index than the one from the industrial polyol. In fact, authors contributing to the literature have already reported that higher values of the hydroxyl index usually lead to an incremented density of the polyurethane foams [77,78]. This is because a high hydroxyl number provides a higher degree of crosslinking of the PUF [79].

In comparison to that, the addition of the flame retardants into the PUF formulation did show a significant effect on the apparent density. It has been said that the presence of this type of additives in a PUF formulation influences the values of this parameter [80]. The most evident difference was detected after the addition of ammonium polyphosphate (APP), as this formulation (F_APP) yielded the PUF with the highest apparent density. Indeed, the obtained value for this parameter was almost twofold that of F_REF and F_IND. The reason for this characteristic was associated with the fact that ammonium polyphosphate could act as a nucleating agent during the formation of polyurethane foams [81]. Therefore, it can get embedded within the foam wall, owing to its low particle size, leading to the formation of both covalent bonds and ionic interactions and resulting in high density values [81]. This fact has been confirmed in other studies, in which the incorporation of APP into a PUF formulation also showed an increase in the apparent density [82,83]. Furthermore, the acid character of APP could result in a lower foaming ratio. In fact, it was presented by Maillard et al., 2020 [84], that the pH has a significant influence during the foaming reaction. They found that an acidic pH can result in the slowing down of the foaming process. Accordingly, the presence of APP in the corresponding formulation and its acidic nature could have slightly slowed down the foaming process, leading to a foam with a lower foaming ratio (less volume). In contrast to this, the incorporation of poyloligomeric silsexquioxane (POSS) in the polyurethane foam showed the lowest value of apparent density. Normally, the addition of this type of filler displays an opposite tendency, as presented in previous research [85,86]. However, it has also been reported that the introduction of some amount of solvent for the dissolution of POSS could lead to a decrease in the final polyurethane foam’s density [87]. Consequently, the previous dissolution of POSS in ethanol carried out in the present work might have caused this decreasing effect in the apparent density of F_POSS.

The mechanical properties of the PUF formulations were also assessed by means of compressive strength analysis. Thereby, the values of compressive stress at 10% strain were recorded as compressive strength and that of compressive modulus was registered as the slope from the curve in the elastic region (Figure 6a,b). It is well known and it has been previously described that the mechanical properties of polyurethane foams are considerably dependent on the apparent density [50,88]. Considering the formulations elaborated in this work, this mentioned behavior was mostly confirmed, based on the results obtained. Thus, the F_APP formulation, which was the one displaying the highest value of density, presented the best mechanical properties. On the contrary, the polyurethane foam with POSS, which had the lowest apparent density of all the formulations, showed the lowest values of compressive strength and modulus. This might also be attributed to the reaction of the additive with the isocyanate. In this case, the crosslinking density of the PUF would be decreased, and thereby, the displayed mechanical properties could be degraded [29]. Regarding the F_REF formulation, the values of the compressive strength and modulus were between those of F_APP and F_POSS, sharing the same tendency in terms of the apparent density.

Concerning the cellular morphology of the PUFs, it was studied through micrographs obtained from the SEM analysis. This analysis was useful to correlate the cellular structure with the results previously displayed concerning the density and mechanical properties of the PUFs.

In Figure 7, it can be seen that in general, the PUFs’ morphology showed a honeycomb structure and closed cells with a homogeneous distribution of the pore size. This can be more evidently observed in the F_REF and F_IND formulations. In contrast to that, formulations with the flame-retardant additives, i.e., F_POSS and F_APP, show a more irregular distribution of the pores, which were filled with the mentioned additives. Moreover, in some regions, they presented a certain level of pore cracking that might be due to filler agglomeration.

Regarding the average pore size and distribution, it was seen that the nature of the polyol did display a significant role. Thus, in the foam formulations that were prepared with the biopolyol, the average size was significantly smaller (306.71 μm, 167.40 μm and 155.28 μm for F_REF, F_POSS and F_APP, respectively) than in the formulation based on the industrial polyol (636.70 μm for F_IND). This could be associated with the fact that the biopolyol presented a higher hydroxyl value than the industrial polyol did. Thereby, a polyol with a higher hydroxyl value would lead to the formation of a PUF with a higher crosslinking density and therefore a smaller cell size [78].

Moreover, other authors have also reported that a polyol with a lower viscosity would result in a larger cell size in the PUF [50]. This is also in agreement with the results obtained here, since the biopolyol had a higher viscosity compared to the industrial one (Table 5).

Another parameter with a considerable influence on the size of the cells was the presence of a filler within the PUFs. It was observed that the introduction of the flame-retardant additives into the formulations produced a decrease in the average size of the cells (Figure 7b,c). The mentioned behavior was also presented by Zhang et al., 2021 [89], who reported that the introduction of SiO_2_ in a rigid polyurethane foam formulation lead to a much smaller cell size than the one of the parent foam with the filler.

These previous observations, extracted from the SEM analysis, were consistent with the results obtained for the density and mechanical properties of the foams. Hence, the fact that F_APP presented the lowest cell size compared to the other formulations would be consistent with it having the biggest density values and the highest mechanical performance. In addition to that, the pores being filled with the APP would also be an explanation of this performance (Appendix A). With respect to the F_POSS, it is true that it displayed a pore size that was similar to that of F_APP; nevertheless, their values of mechanical properties were significantly lower. This characteristic was related in part to the fact that POSS was more difficult to incorporate into the polyol formulation for the development of the foams (in fact, it had to be predissolved into ethanol before being incorporated). This may have caused an irregular distribution of the POSS filler during the crosslinking of the polyurethane foam. Thereby, some areas could present an agglomeration of the particles of POSS, causing the rupture of some of the foam cells. In fact, this situation was confirmed in some of the SEM images, which can be seen in Appendix A from the Appendix A document. In the end, the breakage of some of the cells in the foams could lead to poorer mechanical properties for the F_POSS compared to those of F_APP.

### 3.5. Flame Retardancy and Fireproofing Properties

Concerning the performance of the different foam formulations against fire degradation, three analyses were carried out, namely, UL-94 vertical burning analysis, limited oxygen index test and fire reaction of construction materials procedure. The former two are the most typically used tests within the literature for the evaluation of the fireproofing properties of any material, whereas the latter one is more specific to this type of material at the industrial scale.

Regarding UL-94 vertical burning and LOI analyses, the results obtained are shown in Table 7. On the one hand, from the UL-94 test, it was confirmed that the introduction of both flame retardants (APP and POSS) displayed a significant effect, improving the performance of the PUFs. In this test, the samples were cut to dimensions of 100 × 10 × 10 mm, and they were subjected to two periods of burning of 10 s. The results showed a reduction in the t_1_ and t_2_ from F_REF to the F_POSS and F_APP formulations of between 80 and 90%. Especially in the case of F_POSS, it was observed that during the second burning time, the samples were unable to ignite. This was related to the formation of a formation of a swelled char layer, due to the intumescent effect that was achieved by POSS (Appendix A). It confirmed that this flame retardant was properly embedded within the PUFs, and that can provide a protecting intumescent effect (as seen in the previous test). In addition to that, it should be remarked that both formulations with flame retardants (F_POSS and F_APP) achieved a V-0 rating, whereas the F_REF sample remained on a V-2 rating. Regarding the literature, the results achieved with these flame retardants were comparable to or higher than those presented in other works. For instance, Xu et al., 2022 [90], showed that a commercial PUF with 15% of APP loading only reached a V-1 rating. In other works, using phosphorous and silicon-based flame retardants for PUFs, they showed that it is sometimes necessary to either use a synergist or higher loadings to reach a V-0 rating [91,92]. Compared to F_IND, both F_POSS and F_APP achieved a similar degree of performance. The differences between their t_1_ and t_2_ were low, and so was the divergence found between the lengths reached by the flames. Moreover, all three reached the same rating. In the Appendix A (Appendix A), the final state of the samples after the test is displayed.

On the other hand, the LOI test also confirmed that the incorporation of the flame retardants in the PUF formulations achieved an enhancement of their fireproofing properties. Here, the samples employed were also cut to the same dimensions (100 × 10 × 10 mm). It can be seen that the F_REF formulation reached ≈20% LOI, which is within the range presented by other works in the literature [93,94]. In comparison to F_REF, the F_POSS and F_APP formulations incremented the LOI percentage by ≈10% and 12.5%, respectively. Thereby, F_APP showed the highest performance of all the PUFs based on the biopolyol. In another work by Li et al., 2020 [95], a similar value of LOI was achieved for a rigid polyurethane foam with a similar percentage of additive. In comparison to F_IND, the values obtained by F_POSS and F_APP were slightly lower. In the case of APP for instance, this could be related to the fact that this flame retardant provides a quenching effect during the middle and later stages of combustion. In fact, APP starts its degradation at a point at which half of the PUF is already degraded [28]. In contrast, in the LOI test, it would be more convenient with a flame retardant that would provide a quenching effect in the early stage of the combustion [96].

In the last analysis (Reaction to fire tests—Ignitability of products subjected to direct impingement of flame), the samples were cut to dimensions of 125 × 45 × 25 mm and were subjected to a flame at an angle of 45° for 30 s. Then, different parameters were determined, namely, the percentage of mass remaining after analysis, length burnt, dripping particles and flammable droplets. These results were collected, and the average values are shown in Table 8. It can be observed that between the formulations derived from a biosourced polyol, there was a clear difference when a fireproofing additive was incorporated. In fact, both compounds are well known for being efficient in the protection against fire. On the one hand, polyhedral oligomeric silsesquioxane (POSS) is a silicon-based fireproofing compound with certain functionalities in its terminal position, which can allow for the formation of covalent bonds to the polymeric backbone [54]. On the other hand, APP is an amino-phosphorylated polymer that is widely used lately, owing to its low toxicity and efficiency as a fireproofing compound [97]. Both compounds achieve fire protection by forming a char layer that hinders the combustion of the polymeric material. For instance, it has been proven that a phenyl containing POSS (the one used in this work) is capable of building a char layer of great strength upon contact with fire [87].

With respect to the mass percentage remaining after the analysis, the value obtained for F_REF formulation was low (45.89%), whereis it was incremented to 77% for F_POSS and F_APP after the corresponding flame retardants were added. In addition to that, it was observed that the incorporation of POSS and APP into the PUFs avoided the dripping effect which was occurring in the F_REF formulation. Both enhancements were associated with the creation of the previously mentioned char layer on the PUF surface upon fire contact. Although both formulations contained the lignocellulosic biopolyol, which can act as a carbon source, this charring effect was most significant in the case of F_POSS compared to F_APP. This was because POSS has benzene rings within its structure, which represent an extra carbon source. In contrast to that, APP is lacking this extra carbon source, and this is less favorable for promoting the charring effect during combustion [28]. Moreover, the performance of these biobased formulations compared to the industrial one was at a similar level. Regarding the length of the samples burnt, it was seen that in contrast to the F_IND formulation, in the rest of the cases, the flames covered the whole sample. However, it was noticed that the integrity of the PUF was not significantly compromised, and the inner part of the samples remained untouched (Figure 8). In this sense, a different behavior could be detected in the fire protection mechanisms of F_POSS, F_APP and F_IND. The former formulation presented a clear intumescent effect, which resulted in a swelled-char layer. Other authors have also confirmed this effect of POSS on polymeric matrices [89,98,99]. Nevertheless, in the other formulations, no swelling behavior was observed. Instead, a thin but robust layer was formed.

### 3.6. Thermal Insulating Properties

Thermal conductivity is a property of PUFs with a considerable significance, since it is an indicator of the performance of these materials in terms of thermal insulation applications [100]. Ideally, lower values of this parameter can lead to a higher thermal insulation efficiency. Nevertheless, there are different factors which display a substantial influence over this property, e.g., the blow agent used in the formulation, the density of the foam, the fillers added, the temperature or the humidity [101]. In general, it has been reported that the thermal conductivity of insulating materials used in the industry can be up to 0.05 W·m^−1^·K^−1^ [102]. More specifically, that of PUF foams used in the building and construction sector is in the range of 0.02–0.03 W·m^−1^·K^−1^ [103]. In this work, the performance of the different PUF formulations as thermal insulators was studied, as presented in Figure 9.

Based on the results obtained, it was observed that all the PUF formulations that were developed by employing the biosourced polyol displayed values within a small range (0.02371–0.02149 W·m^−1^·K^−1^). Moreover, these values were similar or even lower to that obtained for the industrial formulation. In addition to that, compared to other works from the literature, in which biobased polyols are introduced into the PUF formulation either partially or completely, the performance obtained here was on average higher [29,57,104].

Regarding the slight differences detected in the values of the thermal conductivity of the foam formulations, they do not appear to be strictly proportional to the density values. This could be especially observed for the F_APP foams, which showed one of the lowest values for the thermal conductivity, despite of having the highest density of all formulations. One reason for that can be that the thermal conductivity does not increase proportionally to the density, and therefore, the values usually vary slightly in the range of density between 30 and 1000 kg·m^−3^ [105]. In addition to that, it has been reported that the thermal conductivity is not only influenced by the density but by other structural parameters such as the cell size and thickness or the strut diameter. Accordingly, the experimental investigation of the thermal conductivity on polyurethane foams is a complex and arduous task [106]. In this case, the lower value of the thermal conductivity might be due more prominently to the addition of fillers in the formulations, namely, flame retardants (POSS and APP). These additives, which have led to a lower cell size of the polyurethane foams, could have resulted in a lower thermal conductivity [107].

## 4. Conclusions

In this work, the synthesis of a polyol derived from a biomass residue at a larger scale was intended, aiming ultimately for the production of PUFs with a total substitution of the mentioned polyol and with a performance that is similar to that of industrial PUF. Considering the results of the previous analyses, several findings could be drawn.

Concerning the liquefaction process, it was proven that big batches of polyol could be obtained without compromising the amount of the final product. In this sense, it was seen that a competitive yield (≈90%) could be achieved by using milder conditions than those employed in other works from the literature. In addition, the biopolyol displayed suitable properties for its utilization in the formulation of polyurethane foams. This could be observed from the properties of the synthesized biopolyols, which in general were similar to those of an industrial counterpart. The main divergences detected were due to the lignocellulosic nature of the biopolyols’ raw material. Nevertheless, they were not significant enough to hinder the subsequent foaming process. In addition to that, it was demonstrated that the biopolyols were compatible with the selected flame retardants without jeopardizing the process of the formation of the polyurethane foams. The structural analysis by FTIR spectroscopy displayed that the flame retardants were properly incorporated into the foams, and TG analysis confirmed that they enhanced their thermal stability.

With respect to the physical and mechanical parameters of the foams, it was corroborated that using a biopolyol resulted in the density falling within the common range used in the industry (30–100 Kg·m^−3^). Additionally, a good level of correlation between the density and the mechanical properties was observed, and these properties of the biopolyol-based formulations were at a similar level to those of the industrial PUF. The study of the morphology also exhibited consistency between the results determined for these two previous properties. One of the most important parts of this study was the analysis of the degradation against fire. In this respect, it was corroborated that the incorporation of the flame-retardant additives significantly improved the performance of the PUF based on the biopolyol. This could be especially observed in the UL94 vertical burning test, in which the formulation with flame retardants improved the rating to V-0, which was the same rating achieved by F_IND. Moreover, a slight increment was also seen in the LOI test after the introduction of the flame retardants in the biobased formulations of PUFs. Additionally, from the results obtained, it could be inferred that POSS was mainly exerting an intumescent effect on the PUF formulation, whereas APP was providing effects in the gas and condensed phases.

Another point of significant importance was the application as a thermal insulator. In this sense, it could be observed that the morphology and pore size influenced the thermal conductivity values obtained (0.02371–0.02149 W·m^−1^·K^−1^). As seen from the results for the different formulations, the different additives, e.g., flame retardants, exerted an effect over this parameter. In general, biobased polyurethane formulations proved their good performance as thermal insulators, with values at the same level or even below that of the industrial formulation.

Considering these results, it could be inferred that the biopolyol-based PUFs developed here could be a good and greener alternative to the commercially available PUFs that are available in the market, with a special mention to their fireproofing properties and to their role as a thermal insulator.

## Figures and Tables

**Figure 1 polymers-16-00258-f001:**
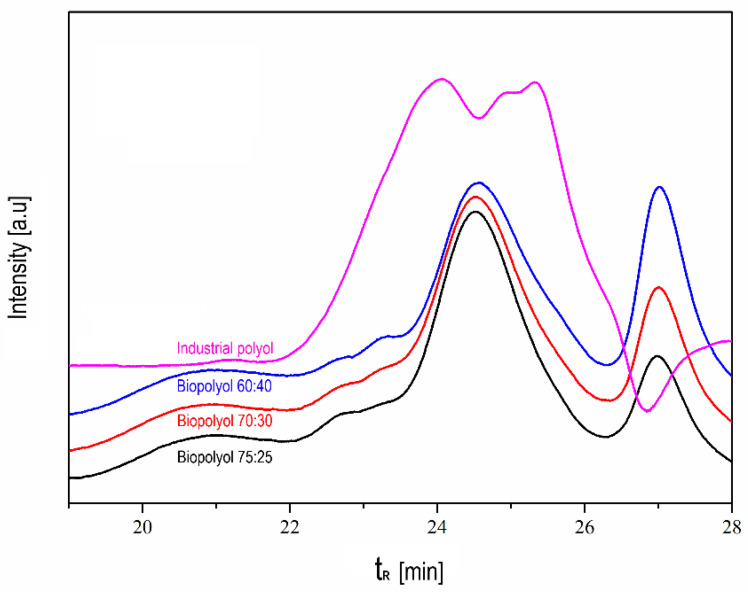
Curves from the size exclusion chromatography analysis for the different biopolyols and the industrial polyol.

**Figure 2 polymers-16-00258-f002:**
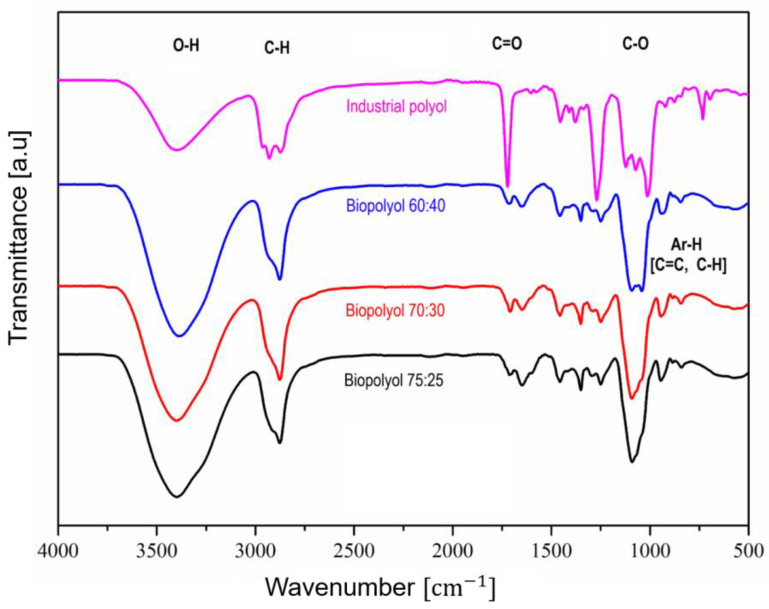
Curves from Fourier Transformed Infrared Spectrometry analysis for the different biopolyols and the industrial polyol.

**Figure 3 polymers-16-00258-f003:**
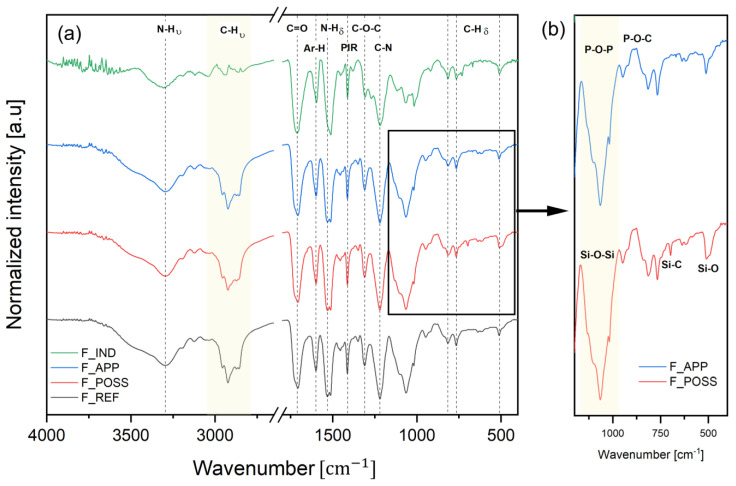
FTIR spectra of the different PUF formulations. (**a**) Whole range of spectra (4000–400); (**b**) Zoom for range 1500–400 to related to the signals of the flame retardants.

**Figure 4 polymers-16-00258-f004:**
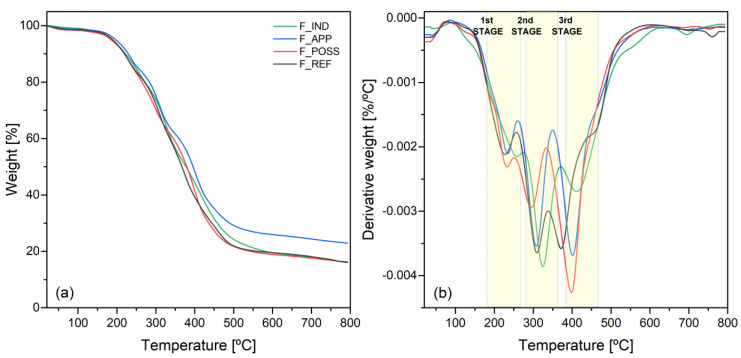
Thermograms of the formulations of PUF: (**a**) thermogravimetric curves and (**b**) derivative thermogravimetric curves.

**Figure 5 polymers-16-00258-f005:**
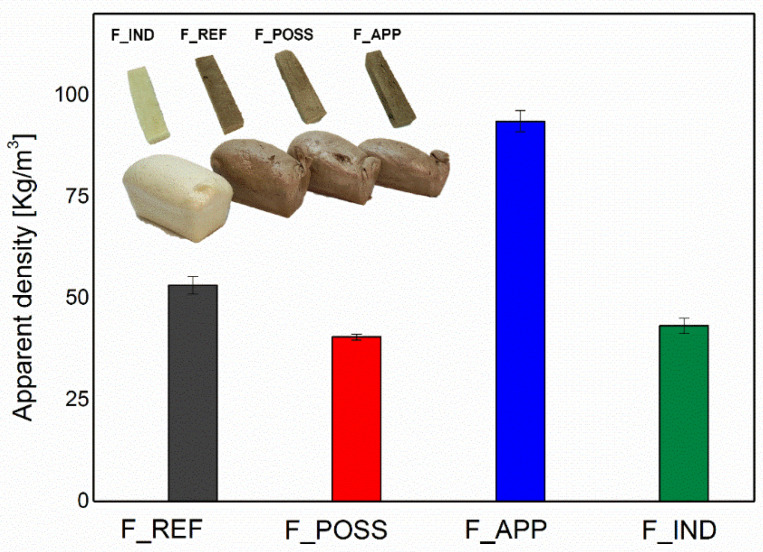
Effect of the different formulations on the apparent densities of polyurethane foams.

**Figure 6 polymers-16-00258-f006:**
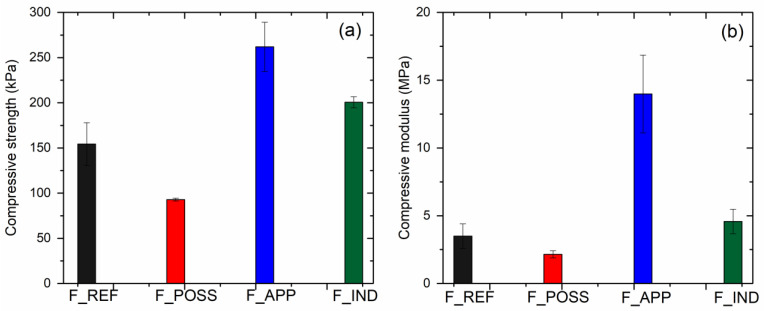
Mechanical properties of the PUF formulation from compressive strength analysis: (**a**) compressive strength at 10% strain and (**b**) slope of the curve in the elastic region.

**Figure 7 polymers-16-00258-f007:**
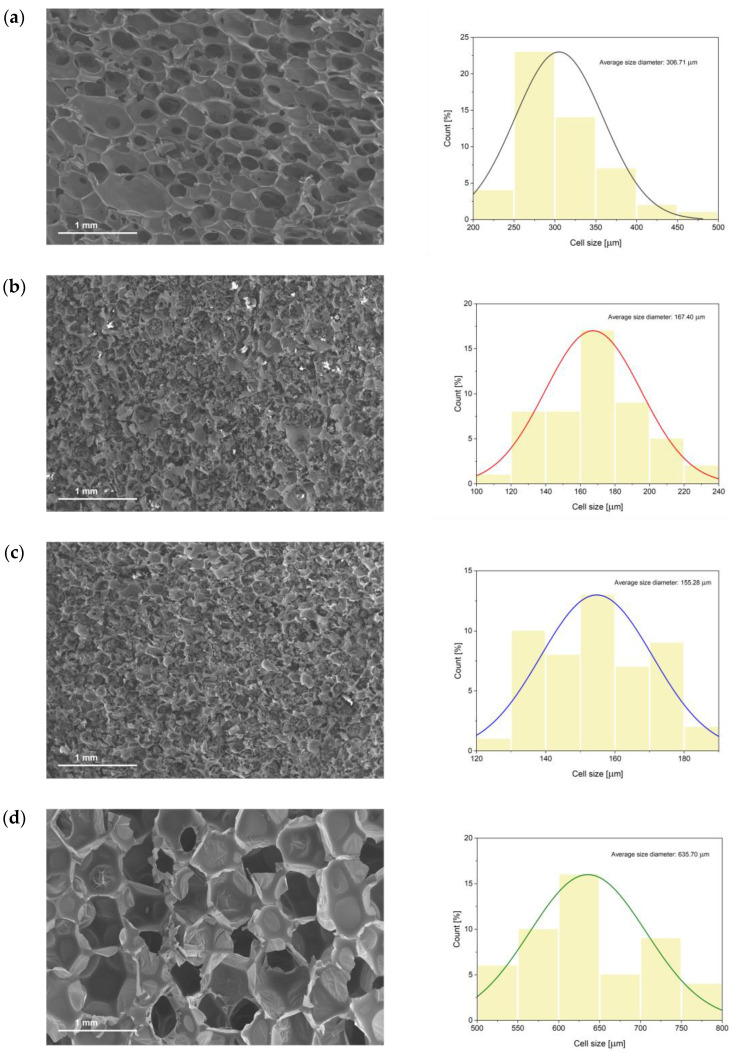
SEM micrographs of different foams and the measure of the pore size distribution: (**a**) F_REF, (**b**) F_POSS, (**c**) F_APP and (**d**) F_IND.

**Figure 8 polymers-16-00258-f008:**
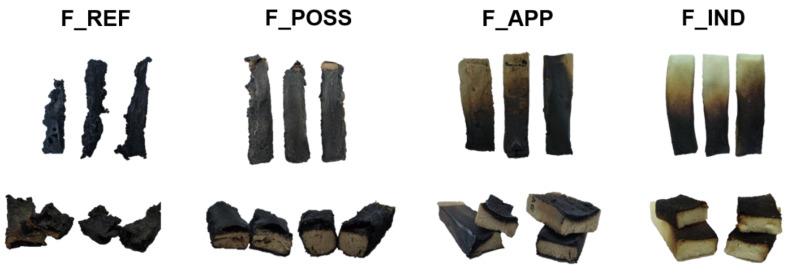
Samples of the different PUF formulations after the flammability tests.

**Figure 9 polymers-16-00258-f009:**
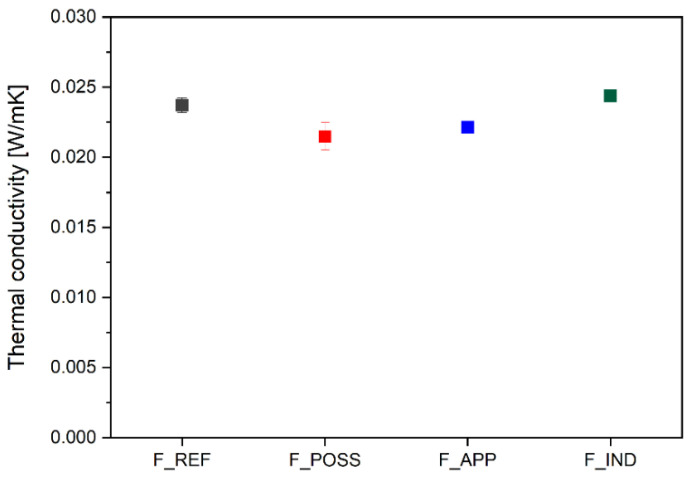
Values of the thermal conductivity for the different polyurethane foam formulations.

**Table 1 polymers-16-00258-t001:** Content of the biobased polyurethane foam formulations prepared.

Formulation	Phase A (pbw ^a^)							Phase B (pbw)
	TEGOSTAB 847100	Biopolyol	DBTDL	POLYCAT 5	BA ^b^	POSS	APP	Isocyanate
F_REF	20	100	16	2	1	---	---	120
F_POSS						12.5	---	
F_APP						---	12.5	

^a^ pwb: parts by weight of biopolyol, ^b^ BA: blowing agent (water).

**Table 2 polymers-16-00258-t002:** Analysis of the raw material.

Chemical Composition of Raw Material (%)				
Cellulose	Hemicellulose	Lignin	Extractives	Ashes
36.36 ± 0.06	16.17 ± 0.16	27.60 ± 0.34	5.84 ± 0.24	0.22 ± 0.11

**Table 3 polymers-16-00258-t003:** Properties of the different biopolyols obtained and the yield of conversion of liquefaction.

Biopolyols	Density (g∙cm^−3^)	Viscosity (mPa∙s)	I_OH_ ^a^ (mg KOH∙g^−1^)	Functionality	Conversion of Liquefaction (%)
75:25	1.15 ± 0.01	394.20 ± 21.49	295.47 ± 46.71	2.74 ± 0.54	78.97 ± 4.35
70:30	1.17 ± 0.01	438.75 ± 22.98	360.29 ± 27.73	3.23 ± 0.25	81.02 ± 0.54
60:40	1.19 ± 0.01	578.3 ± 37.6	546.6 ± 22.32	3.30 ± 0.13	89.15 ± 3.68

^a^ I_OH_: hydroxyl index.

**Table 4 polymers-16-00258-t004:** Molecular weights of the biopolyols obtained during optimization.

Polyols	Molecular Weights (g∙mol^−1^)			Equivalent Weight(g∙mol^−1^)
	M_w_	M_n_	Polydispersity Index (PI)	
BP-75:25	3148	520	5.55	189.87 ± 10.78
BP-70:30	2887	501	6.28	155.71 ± 6.71
BP-60:40	2269	339	7.32	102.63 ± 5.11
Industrial	857	524	1.6	280.5

**Table 5 polymers-16-00258-t005:** Physical–chemical and structural properties of the polyols.

Polyols	Density (g∙cm^−3^)	Viscosity (mPa∙s)	I_OH_ ^a^ (mg KOH∙g^−1^)	Functionality	Equivalent Weight (g∙mol^−1^)	Molecular Weights (g∙mol^−1^)		
						M_w_	M_n_	PI ^b^
BP	1.2 ± 0	578.3 ± 37.6	546.6 ± 22.3	3.3 ± 0.1	102.6 ± 5.1	2269	339	7.3
IP	1.1	300–500	200	1.9	280.5	857	524	1.6

^a^ I_OH_: hydroxyl index, ^b^ PI: polydispersity index.

**Table 6 polymers-16-00258-t006:** Main TGA parameters of the different formulations.

Formulation	T_5%_ (°C)	T_50%_ (°C)	T_max_ (°C)	Residue (%)
F_REF ^a^	189.7	367.0	309.2	16.1
F_POSS ^b^	185.4	379.7	397.8	16.2
F_APP ^c^	197.3	396.0	401.2	22.9
F_IND ^d^	187.9	377.4	324.4	16.3

^a^ F_REF: formulation based on biopolyol and isocyanate without flame retardants, ^b^ F_POSS: formulation based on biopolyol and isocyanate with POSS flame retardant, ^c^ F_APP: formulation based on biopolyol and isocyanate with APP flame retardant, and ^d^ F_IND: formulation based on industrial polyol and isocyanate.

**Table 7 polymers-16-00258-t007:** Parameters for the assessment of the fire degradation of the samples at laboratory scale.

Samples	UL-94 Vertical Burning Test						LOI [%]
	t_1_ [s]	t_2_ [s]	Length Burnt [mm]	Dripping Particles	Flammable Droplets	Rating	
F_REF	27.0 ± 3.5	--- *	100 ± 0	Yes	Yes	V-2	19.9
F_POSS	6.4 ± 2.6	0	100 ± 0	No	No	V-0	21.7
F_APP	3.3 ± 1.3	1 ± 0	100 ± 0	No	No	V-0	22.4
F_IND	4.8 ± 1.3	1 ± 0	97.4 ± 2.2	No	No	V-0	23.8

* F_REF samples were completely burnt and decomposed after the first period of 10 s burning.

**Table 8 polymers-16-00258-t008:** Parameters for the assessment of the fire degradation of the samples at industrial scale.

Formulation	Mass Remaining after Analysis ^a^ [%]	Length Burnt [mm]	Dripping Particles	Flammable Droplets
F_REF	45.9 ± 0.9	125 ± 0	Yes	Yes
F_POSS	79.2 ± 2.1	125 ± 0	No	No
F_APP	79.8 ± 2.6	125 ± 0	No	No
F_IND	91.3 ± 1.6	63.8 ± 4.2	No	No

^a^ Ratio of the sample mass before and after the analysis.

## Data Availability

Data are contained within the article and Appendix A.

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
