# Peer review of "Elaboration of Thermally Performing Polyurethane Foams, Based on Biopolyols, with Thermal Insulating Applications"

_polymers, 2024, doi:10.3390/polym16020258_

Round 1
Reviewer 1 Report
Comments and Suggestions for Authors
The work concerns the preparation and use of polyol obtained from natural raw materials (wood sawdust) for the production of rigid polyurethane foams. Conditions for optimizing the synthesis of this polyol were developed, its properties and the properties of rigid polyurethane foams modified with flame retardants obtained from it were examined. The work falls within the scope of new trends aimed at replacing polyols obtained from petrochemical raw materials with natural raw materials.
It should be published after major corrections in accordance with the comments below.
1. In the INTRODUCTION, it would be beneficial to more precisely discuss the literature on polyols obtained from wood waste and sawdust.
2. Please provide the chemical name of the POLYCAT 5 catalyst
3. Please characterize the structure of industrial polyol (IP).
4. Lines 155-158. How to fit 6 liters of PEG/G mixture + 600 g of sawdust into a 6 liter flask?
5. Please introduce SI.1.1 "Optimization of the liquefaction process-ratio of solvents" from SUPPLEMENTARY INFORMATION into the main part of the article
6. Please describe in the main part of the article how the composition of the foamed composition was optimized (Table 1), i.e. please enter point SI.1.4 "Development of the PUFs formulations: optimization process" from SUPPLEMENTARY INFORMATION
7. Please move subsection SI2.1 "Structure and performance of the polyol during the optimization of the liquefaction process" from SUPPLEMENTARY INFORMATION together with tables SI1 and SI2 without Figure SI3 to the RESULTS AND DISCUSSION chapter.
8. Similarly, the IR spectra from Figure SI4 should be transferred to the main part of the article along with their interpretation.
In the SUPPLEMENTARY INFORMATION, Figure SI4 was incorrectly signed as chromatography analysis
9. Based on the LOI and length burnt values, it is difficult to assess the use of the flame retardants used as factors that significantly reduce burning. Moreover, POSS flame retardants are very expensive. In general, some improvement in flame retardant properties was observed compared to the F-REF foam, while the flame retardant properties of F_POSS and F_APP foams were worse than those of F_IND foam.
Did the F_IND foam contain any flame retardants?
10. The study lacks tests such as water absorption and polymerization shrinkage for all obtained polyurethane foams, as well as the influence of the flame retardants used on these properties. These properties are important when determining the direction of application of the foams.
11. In fact, the information presented in the CONCLUSIONS section is not a conclusion, but a summary of the work and should be titled as such.
Author Response
Dear reviewer,
The authors values the points raised to the previous version of our manucript and we have tried to answer them, so that the quality of our work could be improved. Please find attached the response to each of the questions pointed out in the review.
Regards,

Reviewer 2 Report
Comments and Suggestions for Authors
The article analysis biopolyurethane foams produced from wood sawdust waste-based biopolyol. The resulting product's application area is thermal insulatin, therefore, I have few remarks which I hope will help to improve the quality of the article.
1. The title states that the application of the developed foams is thermal insulation. However, no results of closed cell content are presented. In addition, ageing procedure of thermal insulating polyurethane foams also has to be carried out.
2. Not all necessary flammability properties are covered such as LOI, cone calorimetric measurements, DMA.
3. What about SEM images of char residues after flammability test? Or Raman spectra?
Author Response

(The authors gave the same response as above.)

Reviewer 3 Report
Comments and Suggestions for Authors
In this study, the author focuses on the synthesis of biopolyols that have the potential to compete with commercial polyols and be utilized in the production of bio-based rigid polyurethane foam (PUF). Moreover, various formulations of bio-based rigid polyurethane foam (PUF) were developed to evaluate the impact of two flame retardants, specifically polyhedral oligomeric silsesquioxane (POSS) and amino polyphosphate (APP), on their thermal characteristics, degradation, and fireproofing mechanism. Overall, I would like to suggest that this work be published in MDPI Polymers after addressing the following issues.
1) The author must mention the full form of the designated formulation in the preparation of the polyurethane foam part in order to elucidate Table # 4 for readers.
2) Why does the addition of ammonium phosphate (APP) cause to increase in the density of PUF? The author must elaborate in the main manuscript.
3) Figure 5 (b, c) exhibits similar average pore size, so why does F_POSS show lower mechanical properties than F_APP?
4) Concerning the effect of flame retardant while assessing the LOI test, the F_IND PUF specimen shows a higher value of LOI than the F_POSS and F_APP. Why? What factors lead to the reduction of the LOI even though the addition of flame retardant in F_POSS and F_APP in comparison with F_IND without flame retardant?
5) The F_APP specimen exhibits lower thermal conductivity despite showing higher apparent density. Is thermal conductivity independent of density? The author must explain the reason in the main manuscript.
Comments on the Quality of English Language
No problem
Author Response

(The authors gave the same response as above.)

Reviewer 4 Report
Comments and Suggestions for Authors
The English should be revised.
The title should be revised as thermal conductivity is only one of the aspects covered in the paper and thermal insulation should be included so as not to confuse it with acoustic insulation, for example.
This referee considers it advisable to include some of the supplementary information on all the diagrams of the manufacturing processes in the manuscript for a better understanding of them.
Table I seems incomplete
It is advisable to indicate in the text the code of the standard and the full name to be included in the list of references.
In section 2.5.3, the equipment used to measure the thermal conductivity of the material according to UNE en 12667 should be specified.
The format of Table II should be improved
Change the title of section 3.6 to Thermal insulating properties.
The order of the figures should be revised
Author Response

(The authors gave the same response as above.)

Round 2
Reviewer 1 Report
Comments and Suggestions for Authors
The work is suitable for publication. All reviewer comments were carefully explained.
Reviewer 2 Report
Comments and Suggestions for Authors
Authors did not include the important test results such as DMA, cone calorimeter and other. Therefore, I remain with the same decision as it was previously. I would strongly suggest seeking collaboration and adding the suggested tests appropriately.